# Regulatory Effects of Coffee/Chlorogenic Acid and Tea/Epigallocatechin-3-*O*-Gallate on microRNA in Association with Their Anticancer Activity

**DOI:** 10.3390/cimb47110898

**Published:** 2025-10-29

**Authors:** Mamoru Isemura, Sumio Hayakawa, Tomokazu Ohishi, Noriyuki Miyoshi, Ryuuta Fukutomi, Yoriyuki Nakamura

**Affiliations:** 1 Tea Science Center, University of Shizuoka, Suruga-ku, Shizuoka 422-8526, Japan; yori.naka222@u-shizuoka-ken.ac.jp; 2Department of Biochemistry and Molecular Biology, Graduate School of Medicine, Nippon Medical School, Bunkyo-ku, Tokyo 113-8602, Japan; hayakawa_sci@icloud.com; 3Laboratory of Oncology, Institute of Microbial Chemistry (BIKAKEN), Microbial Chemistry Research Foundation, Shinagawa-ku, Tokyo 141-0021, Japan; ohishit@bikaken.or.jp; 4 Graduate School of Integrated Pharmaceutical and Nutritional Sciences, University of Shizuoka, Suruga-ku, Shizuoka 422-8526, Japan; fukutomi_ryuuta@mac.com

**Keywords:** coffee chlorogenic acid, tea catechin EGCG, epidemiology, cancer, microRNA, ROS

## Abstract

Epidemiological studies have provided evidence to show that the consumption of coffee and green tea has beneficial effects against cancer. Chlorogenic acid (CGA) in coffee and epigallocatechin-3-*O*-gallate (EGCG) in tea are involved in these effects. Research also suggests that the anticancer effects of coffee and tea may vary depending on the type of cancer, although the reasons for this remain unclear. As bioactive food factors, CGA and EGCG can contribute to epigenetic modification to exert their anticancer activity. One of the anticancer mechanisms is the one associated with reactive oxygen species (ROS). CGA and EGCG possess activities that initiate anticancer pathways by down-regulating ROS and NF-κB, and up-regulating AMPK. CGA and EGCG can regulate non-coding RNAs, including cancer-associated microRNAs. This review provides updated information on how CGA and EGCG exhibit anticancer activity via ROS-dependent anticancer pathways by regulating microRNA expression.

## 1. Introduction

Coffee and tea, made from *Camellia sinensis* (tea plant), are the most consumed beverages worldwide [1]. Intake of these beverages is believed to have beneficial effects in various diseases, including cancer. Chlorogenic acid (CGA) in coffee and epigallocatechin-3-*O*-gallate (EGCG) in tea (Figure 1) have been shown to contribute to these effects, and we have discussed these aspects [2,3,4].

This paper provides updated information, including observational epidemiology of coffee and tea consumption, regulation of microRNA (miR) by CGA and EGCG, and roles of miR in reactive oxygen species (ROS)-mediated anticancer pathways.

## 2. Observational Epidemiology of Coffee and Tea Consumption

This section is divided by subheadings. It will provide a concise and precise description of the experimental results, their interpretation, and the experimental conclusions that can be drawn.

### 2.1. Human Studies on Consumption of Coffee and Tea

Several epidemiological studies have shown the anticancer effects of coffee. A meta-analysis reported in 2020 for observational epidemiological studies on coffee consumption for 26 different cancers involving 364,749 cancer cases provided evidence to show that coffee consumption is inversely associated with cancer risk of endometrium, liver, oral cavity, pharyngeal, and skin cancers [3,5]. In addition, this meta-analysis suggested a reduced risk for mouth, pharynx, larynx, and skin cancers. It may also be inversely associated with the risk of breast, colon, colorectal, esophageal, and nonmelanoma skin cancers.

However, the same analysis showed that higher coffee intake was associated with an increased risk of bladder cancer, childhood acute lymphocytic leukemia, and possibly lung cancer. The additional results of epidemiological studies on coffee consumption reported before 2020 are available in a previous review [3].

There have been many observational epidemiological studies on tea consumption as well. For example, a survey in 2013 for prospective cohort and case–controlled studies revealed that green tea consumption showed risk-reducing effects across a total of 39 cancer sites, including breast, colon, esophagus, kidney/bladder, lung, ovary, pancreas, prostate, and stomach cancers, whereas 46 cases showed no risk reduction [6]. In the case of black tea, 28 and 92 cases showed risk reduction and no risk reduction, respectively, for these cancers [6]. These findings suggest that green and black teas have preventive effects in some types of cancer.

Additional information is available based on literature published before 2020 in our previous review, which also contains comparisons between coffee and tea [3].

Here, updated information is presented in Table 1. This adds evidence to show the favorable effects of green tea consumption on some types of cancer, such as breast, colon, lung, and blood cancers, and coffee consumption on liver, endometrial, and skin cancers. Coffee consumption may be related to the higher risk of bladder cancer. In contrast, tea consumption is related to an increased risk of skin cancer, while daily coffee consumption is a protective factor [7]. Tea consumption was reported to be associated insignificantly with an increased risk for thyroid cancer [8]. Thus, green tea and coffee are likely to have some differences in site-specific anticancer effects.

When creating a correlation plot using country-level coffee consumption and age-standardized cancer incidence based on the data of the FAO/European Coffee Report 2023–2024 (https://www.ecf-coffee.org/european-coffee-report-2023-2024/ accessed on 22 October 2025) and age-standardized cancer incidence data (excluding non-melanoma skin cancer) from WHO/IARC GLOBOCAN 2022 (https://www.wcrf.org/preventing-cancer/cancer-statistics/global-cancer-data-by-country/#global-cancer-incidence-both-sexes accessed on 22 October 2025), the results revealed a weak positive association between per capita coffee consumption and total cancer incidence, indicating that coffee intake alone does not explain global variation in cancer occurrence (Figure 2). However, a meta-analysis of 59 cohort studies reported that higher coffee intake was associated with a reduced overall cancer risk (relative risk (RR) = 0.87, 95% confidence interval (CI) = 0.82–0.92), and specifically lower risks of liver, endometrial, and prostate cancers, whereas an increased risk was observed for lung cancer (RR = 2.18, CI: 1.26–3.78) [106]. A meta-analysis review [82] concluded that coffee consumption shows highly suggestive inverse associations with liver and endometrial cancer risks, suggesting potential protective effects against certain cancer types rather than overall carcinogenicity.

Some studies have found conflicting results regarding the effects of coffee and tea consumption. These discrepancies may stem from differences in confounding factors such as the extraction method, degree of roasting, coffee species, serving temperature, method of quantifying consumption, beverage temperature, acrylamide contents, alcohol consumption, cigarette smoking status and healthcare accessibility, and differences in genetic and environmental factors such as sex, race, and age, lifestyle, genetic polymorphisms, and intestinal microbiota [107,108]. It is expected that future studies will clarify the reason for the inconsistent results and may provide clues to establish the anticancer effects of these beverage intakes.

A recent systematic review and meta-analysis by Zhang et al. [109] found that green tea/EGCG consumption reduced cancer risk with statistical significance compared to controls. These authors noted risk reduction in prostate cancer (RR = 0.43, CI = 0.22–0.83), oral cancer (RR = 0.44, CI = 0.01–0.87), gallbladder cancer (RR = 0.72, CI = 0.51–0.94), and hematological cancers (RR = 0.72, CI = 0.49–0.95), suggesting that green tea or EGCG intake may prevent some types of cancer.

### 2.2. Human Clinical Intervention Studies on Consumption of Coffee and Tea

Human intervention studies on coffee/CGA consumption are scarce. Kang et al. [110] conducted a phase I trial for CGA in injectable form in patients with recurrent high-grade glioma. The results indicated that CGA has a favorable safety profile and provides certain clinical benefits to patients with high-grade glioma relapsing following standard therapies. This grade of CGA would promote studies to test the clinical application of CGA for various diseases, including cancer.

A recent systematic review and dose–response meta-analysis of randomized controlled trials (RCTs) by Samavat et al. [111] found that consumption of green coffee bean extract significantly decreased systolic blood pressure (SBP) (weighted mean difference (WMD) = −2.95 mmHg; CI = −4.27 to −1.62; *p* < 0.001) and diastolic blood pressure (DBP) (WMD = −2.15 mmHg; CI = −2.59 to −1.72; *p* < 0.001). The anticancer effect of coffee consumption awaits verification by future RCTs.

A comprehensive review on CGA by Gupta et al. [112] also pointed out that only a few clinical reports have demonstrated the effectiveness of CGA as a therapeutic agent.

As for tea, habitual tea consumption is generally safe (e.g., 704 mg/day in beverage), but higher doses of EGCG (≥800 mg/day) may cause liver toxicity [113]. Filippini et al. [97] evaluated the results of 11 RCTs on green tea supplementation. In most cases, evidence for the beneficial effect of green tea extract on cancer was insufficient. Adverse effects of green tea extract intake were also reported, suggesting that future RCT studies are required to confirm green tea’s effects [97].

Polyphenon^®^ E is a standardized catechin preparation of green tea extract and is proposed as a topical treatment of genital warts. Its efficacy has been demonstrated in several clinical studies [114]. Genital warts are caused by human papillomaviruses (HPVs), suggesting their possible application to HPV-associated cancers such as cervical cancer and lymphocytic leukemia [114]. A clinical trial showed that the treatment with Polyphenon^®^ E ointment or capsules or both of 51 patients with HPV-infected cervical lesions resulted in an overall 69% response rate as compared with that of 10% in the untreated groups [115]. A review article by Norman et al. [116] reported nine clinical studies with a National Clinical Trial number. Although Sinicrope et al. [117] reported that Polyphenon^®^ E did not significantly reduce the number of rectal aberrant crypt foci, further clinical intervention studies may provide clear evidence of the anticancer effects of green tea.

## 3. Regulatory Effects of Coffee/CGA and Tea/EGCG on miRs

Throughout this section, we specify the concentrations of CGA and EGCG used in each cited study (e.g., 25 µM and 50 µM CGA for miR-20a suppression, 10–60 µM EGCG for miR-483-3p downregulation) so that readers can evaluate dose–response relationships.

As bioactive food factors, CGA and EGCG can contribute to epigenetic modification and exert their anticancer activity. For example, hypermethylation or hypomethylation of DNA is closely related to tumorigenesis [118]. Lee and Zhu [119] demonstrated that CGA inhibits the growth of cultured MCF-7 cells through the inhibition of DNA methyltransferase (DNMT), and Pal et al. [120] showed that EGCG reduces the proliferation of HeLa cells and expression of DNMT-1. As shown below, inhibition of DNA methyltransferase would affect the biogenesis of miRs.

EGCG is known to decrease histone deacetylase activity, leading to the increased acetylation levels of histone H3 and H4 [121]. Combining the two findings that ionizing radiation triggers histone modification, such as acetylation of histone H3 and histone H4, leading to upregulation of miR-34a, and that EGCG can upregulate miR-34a, suggests that EGCG-induced histone modifications contribute to the upregulation of miR-34a.

These are the examples to explain how CGA and EGCG can modulate miR expression. EGCG’s ability to bind DNA and RNA [122] represents another possible mechanism. These proposed mechanisms are based on limited in vitro evidence; thus, additional experimental and clinical studies are required to determine whether CGA and EGCG modulate miRNA expression through DNMT inhibition, histone modifications, direct nucleic acid binding, or other yet to be identified pathways.

miRs, which are small single-stranded molecules (ca. 20 to 25 nucleotides), play a role in epigenetic modification involved in tumorigenesis [4,123]. Various dietary polyphenols, including CGA and EGCG, have been shown to regulate miRs as exemplified above and to exert beneficial effects in diseases such as cancer. Several examples have been described in previous papers [4]. Here we provide updated information in Table 2.

CGA and EGCG have been shown to upregulate miR-200c. Davalos et al. [164] found that all cell lines with hypermethylation at CpG islands have significant loss of mature and pri-miRNA expression for the miR-200 family as compared with the CpG unmethylated cell lines derived from the same tumor type, indicating that DNA methylation modulates expression of miR.

Chang et al. [165] have demonstrated that p53 directly induces the expression of miR-34a by promoting transcriptional activity. Since upregulation of p53 by CGA and EGCG is well known [4], these dietary factors are expected to upregulate miR-34a.

Oncogenic miR-17 family miR-20a, miR-93, and miR-106b bind to p21 mRNA and suppress its expression, leading to cancer development [128]. Several studies have demonstrated downregulation of these miRs by CGA and EGCG [4]. Figure 3 illustrates how these miRs downregulate p21 expression.

Table 2 provides an update on studies to show the regulatory effects of CGA and EGCG on miRs. There are some differences in these regulations. This may be due to cancer-specific differences or differences between cell subtypes of the same cancer origin. For example, miR-125b was upregulated in cervical carcinoma SiHa cells, but downregulated in CA33 cells and HeLa cells upon treatment with EGCG [139].

Several studies on EGCG have provided the results of microarray and next-generation sequencing (NGS) [132,135,136,161] (Table 3). Although some of these data have been evaluated by quantitative reverse transcription-polymerase chain reaction (qRT-PCR), the majority remain unconfirmed. Table 3 shows similar results to those based on qRT-PCR, as exemplified by EGCG’s upregulation of miR-34a and let-7a, but there are some conflicting findings. Yamada et al. [163] demonstrated that, in addition to let-7b, both let-7a and let-7e were upregulated in a real-time PCR analysis. Although differences may be due to those in cell types used, caution is warranted regarding data based solely on microarray/NGS analysis.

Thus, NGS platforms provide a large number of findings, but these may not be biologically robust and may lack reproducibility. Therefore, validation of these findings would be necessary. It should be pointed out that similar research on CGA is very scarce, although the reason for this is unclear.

## 4. miR Targets in ROS-Associated Anticancer Pathways

CGA and EGCG are well known to possess both antioxidant and pro-oxidant properties, as discussed previously [2]. However, since the pro-oxidant properties of CGA and EGCG require higher concentrations (e.g., 500 μM EGCG) [166], their radical scavenging activity is thought to primarily contribute to their anticancer activity. Accordingly, the present paper is concerned with the pathways in which CGA and EGCG act as radical scavengers (see Figure 4).

Table 4 and Table 5 summarize how miRs regulate targets in ROS-mediated anticancer pathways. As indicated in Table 2, some miRs undergo cell type-dependent regulation (e.g., upregulation and downregulation) by CGA and EGCG. Table 4 and Table 5 list only mRNAs relevant to the anticancer activity of CGA and/or EGCG along this pathway.

## 5. Involvement of miRs in the Anticancer Pathway Associated with ROS-Scavenging Activities of CGA and EGCG

Table 6 provides updated information that was given previously [2]. It shows a continuing recognition that CGA and EGCG have the activities to down-regulate ROS and nuclear factor-κB (NF-κB), and to up-regulate AMP-activated protein kinase (AMPK) in various biological activities. Based on these data and our previous discussions [2,3,108], Figure 4 depicts a proposed ROS-mediated anticancer pathway in which CGA and EGCG can be involved. Figure 4 also includes information on how miRs, based on the data in Table 4 and Table 5, regulate the components of this pathway.

**Table 6 cimb-47-00898-t006:** Regulatory effects of CGA and EGCG on ROS, AMPK, and NF-κB.

Polyphenols	AMPK Up Stimulation/ Upregulation	ROS Down Suppression/ Downregulation	NF-κB Down Suppression/ Downregulation
CGA	Ping et al. [196] Silva et al. [197] Saadatagah et al. [198]	Wójciak et al. [199] Huimei Chen et al. [200] Sharma et al. [201]	Komeili-Movahhed et al. [202] Negm et al. [203] Lin et al. [204]
EGCG	Peng et al. [205] Tian et al. [206] Wang et al. [207]	Yuan et al. [208] Khan et al. [209] Haoxiang Chen et al. [210]	X. Li et al. [211] Z.-D. Li et al. [212] Zhang et al. [213]

AKT, AKT serine/threonine kinase 1; Bax, Bcl-2 associated X protein; Bcl-2, B-cell lymphoma 2; DDR1, discoidin domain receptor 1; EGFR, epidermal growth factor receptor; ERK, extracellular signal-regulated kinase; OGT, *O*-GlcNAc transferase; KLF4, Kruppel-like factor 4; KRAS, KRAS proto-oncogene; IL, interleukin; MMP, matrix metalloproteinase; mTOR, mammalian target of rapamycin; PAK4, p21-activated kinase 4; PI3K, phosphatidylinositol-3-kinase; PTEN, phosphatase and tensin homolog deleted on chromosome 10; VEGF, vascular endothelial growth factor; VEGFR, vascular endothelial growth factor receptor; Wnt, wingless-related integration site.

## 6. Perspectives

In a recent study investigating the effects of a nutritional supplement on human obesity, Joshua et al. [214] found that consumption of a supplement containing green coffee bean and green tea extracts resulted in significantly reduced plasma levels of miR-34a and miR-122. A similar experimental approach would reveal the efficacy of CGA and EGCG.

In an animal experiment, Kang et al. [158] showed that oral administration of EGCG at a concentration equivalent to daily achievable dosages of tea drinkers suppressed miR483-3p-induced metastasis of hepatocellular carcinoma cells. EGCG induced hypermethylation of the miR483-3p promoter region via epigenetic mechanisms, thereby downregulating miR483-3p expression in these cells. These authors suggested that regular tea consumption can suppress metastasis through downregulation of miR-483-3p, which upregulates vimentin expression and downregulates E-cadherin expression [158], since these events are associated with cancer invasiveness and metastasis [215].

As one of the possible mechanisms, Figure 5 illustrates how exogenous EGCG can suppress metastasis via *O*-GlcNAc transferase (OGT), which is involved as a target of miR-483-3p [195].

One intriguing emerging research area involves analyzing miRs in foods that may exert functional effects after ingestion. Zhang et al. [216] found that plant food-derived miRNAs are present in human and animal sera, and demonstrated that one of these exogenous mature plant miRs is functional. Huang et al. [217] found, based on an NGS analysis, the presence of various miRs as new components of matcha, one of the green tea products. Thus, it would be interesting to examine whether or not food-derived miRs can have beneficial effects in human diseases, including cancer.

A recent comprehensive review by Fujimura et al. [218] pointed out that a variety of biomolecules, such as citrus polyphenols and sulfur-containing food factors, can potentiate EGCG sensing by 67-kD laminin receptor (67LR). Similar synergistic effects may be expected for other polyphenols, including CGA. It would be interesting to examine whether or not such a combination induces any changes in miRs.

## 7. Conclusions

Human observational epidemiological studies have yielded only limited evidence for the reduced cancer risk by consumption of coffee and tea, although CGA and EGCG, major bioactive constituents of these beverages, have been shown to have beneficial effects against various types of cancer. Although a global correlation plot shows a trend of increased risk of total cancer cases with coffee consumption, several epidemiological studies have suggested inverse associations in some specific cancer types. Future studies, including human clinical intervention studies, would be needed to confirm the cancer-preventive effect of these food-derived bioactive factors.

Furthermore, coffee and tea contain a wide range of bioactive constituents besides CGA and EGCG—such as caffeine, diterpenes, trigonelline, melanoidins, and acrylamide—that may individually or synergistically influence cancer risk. Therefore, attributing the observed anticancer effects solely to CGA or EGCG is an oversimplification; direct comparisons using preparations with and without these compounds or 100% purified CGA/EGCG are necessary to clarify their specific contributions. These inconsistencies may also reflect heterogeneity across study populations (e.g., age, sex, smoking prevalence, and genetic background) and differences in the concentrations of bioactive constituents present in various coffee and tea preparations. Variation in dose or exposure level of CGA/EGCG could result in divergent biological effects. Future studies should consider these demographic and compositional variables when interpreting risk estimates.

CGA and EGCG share similar properties in that they can scavenge ROS, which triggers the anticancer pathway leading to cell cycle arrest, apoptosis, and the prevention of inflammation and metastasis. These polyphenols can modulate the expression/activity of multiple components of this pathway by increasing the expression of tumor-suppressing miRs and decreasing the expression of oncogenic miRs in general. Therefore, the effects of these miRs may additively or synergistically enhance the anticancer effects of CGA and EGCG.

EGCG appears to have a specific feature in that it can act via 67LR in anticancer effects and induce miR-let-7b [163]. Although no data are available for CGA, this unique mechanism of action may be one potential explanation for the differences between coffee and tea consumption in cancer-specific effects observed in epidemiological studies. Further studies would clarify the reason for the observed differences in the anticancer effects of the consumption of coffee/CGA and tea/EGCG. In addition, most evidence described in this review remains in vitro or indirect, and future human studies would be needed before firm conclusions can be drawn.

## Figures and Tables

**Figure 1 cimb-47-00898-f001:**
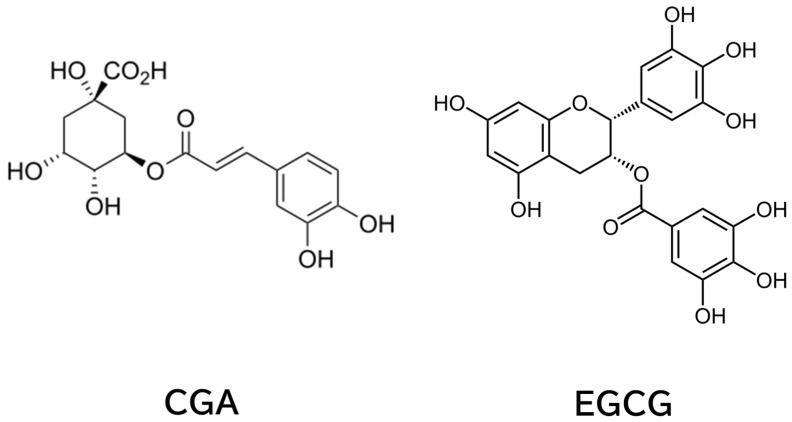
Chemical Structures of CGA and EGCG.

**Figure 2 cimb-47-00898-f002:**
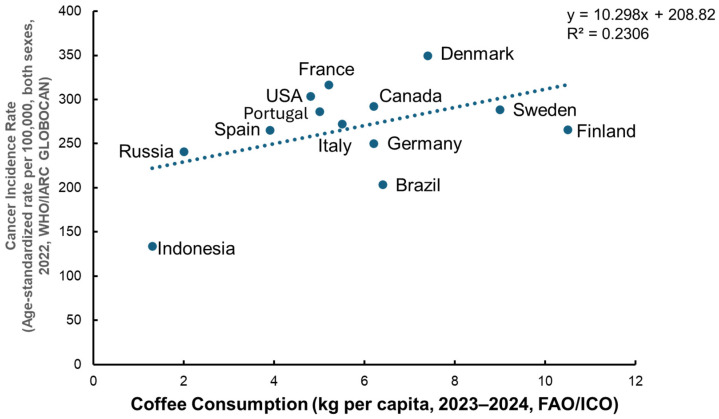
Relationship between coffee consumption and age-standardized total cancer incidence. A correlation analysis was conducted using country-level data on coffee consumption (kg per capita, 2023–2024) from the FAO/International Coffee Organization (ICO) and age-standardized total cancer incidence (per 100,000 population, both sexes, excluding non-melanoma skin cancer, 2022) from WHO/IARC GLOBOCAN. The analysis revealed a weak positive association between per capita coffee consumption and total cancer incidence (R^2^ = 0.23), indicating that coffee intake alone does not explain global variations in cancer occurrence. Data sources: FAO/European Coffee Report 2023–2024 (https://www.ecf-coffee.org/european-coffee-report-2023-2024/, accessed on 22 October 2025); WHO/IARC GLOBOCAN 2022 (https://www.wcrf.org/preventing-cancer/cancer-statistics/global-cancer-data-by-country/#global-cancer-incidence-both-sexes, accessed on 22 October 2025).

**Figure 3 cimb-47-00898-f003:**
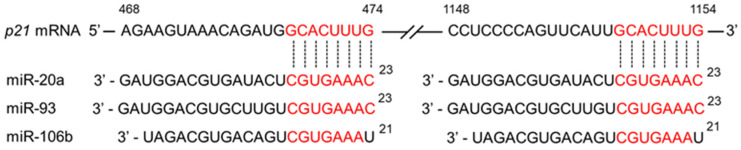
Schematic presentation of the downregulation of p21 expression by miRs. The binding of three miR-17 family miRs (miR-20a, miR-93, miR-106b) to p21 mRNA reduces p21 protein expression. CGA and EGCG may contribute to anticancer effects by suppressing the expression of these miRs, thereby increasing p21 levels.

**Figure 4 cimb-47-00898-f004:**
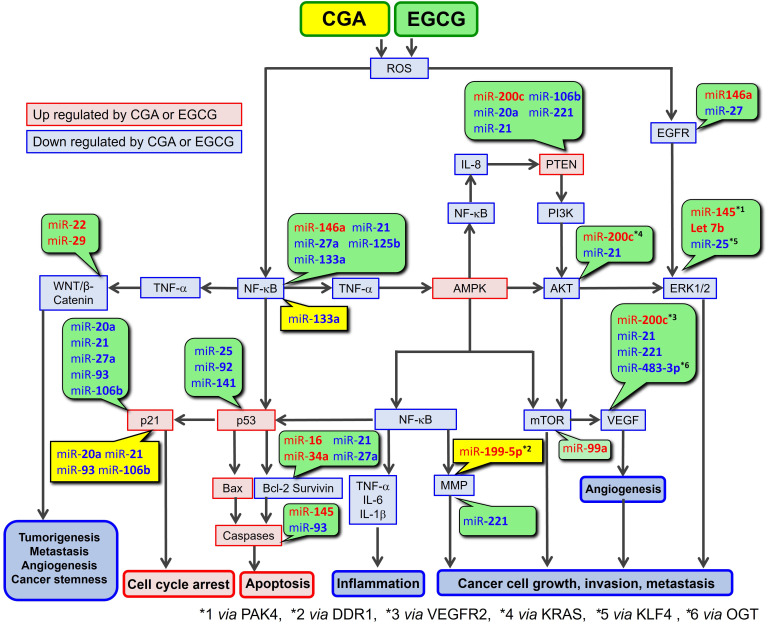
ROS-mediated anticancer pathways involving CGA and EGCG, and microRNA regulation of the components of this pathway. CGA’s and EGCG’s effects on miRs are presented in yellow and green boxes, respectively.

**Figure 5 cimb-47-00898-f005:**
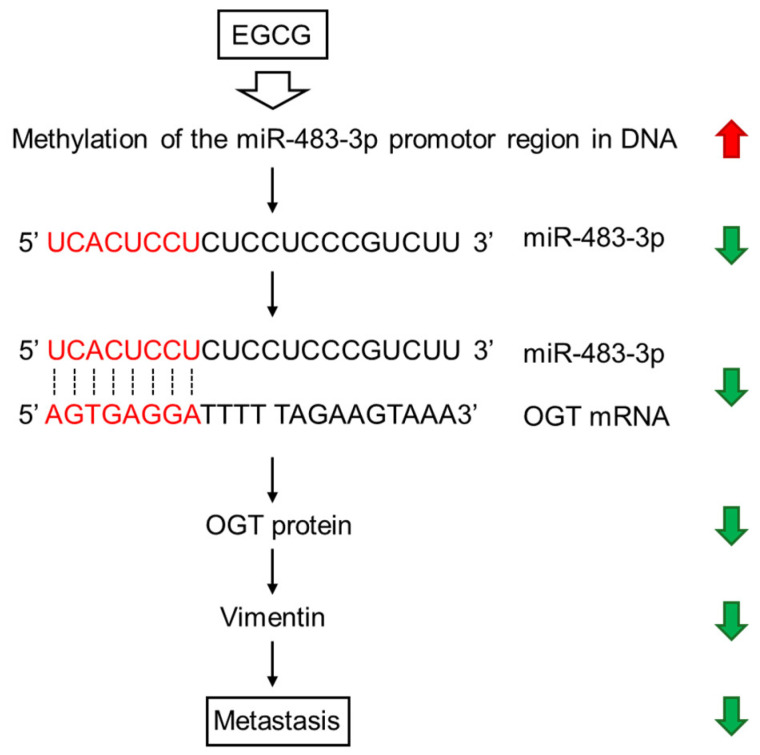
EGCG-induced DNA methylation exerts anticancer effects through downregulation of miR-483-3p, OGT, and vimentin.

**Table 1 cimb-47-00898-t001:** Anticancer effects in an observational epidemiology study of coffee and tea *.

Cancer Type	Coffee/CGA Upregulation	Coffee/CGA Downregulation	Coffee/CGA No Association	Tea/EGCG Upregulation	Tea/EGCG Downregulation	Tea/EGCG No Association
Bladder cancer	Zhao et al. [5] Yu et al. [9] Zhang et al. [10]		Hashemian et al. [11]		Hashemian et al. [11] Zhao et al. [12] Al-Zalabani et al. [13] Zhang et al. [14]	
Blood cancer/ lymphocytic leukemia/childhood acute lymphoblastic leukemia/acute myeloid **	Zhao et al. [5] Milne et al. [15] Karalexi et al. [16] Msallem et al. [17] Flores-García et al. [18] Torres-Duarte et al. [19]	Pranata et al. [20] Malmir et al. [21]	Mirtavoos-Mahyari et al. [22]		Malmir et al. [21] Cote et al. [23] Pranata et al. [20] Zhao et al. [12]	Karalexi et al. [16] Mirtavoos-Mahyari et al. [22] Milne et al. [15]
Brain cancer/glioma	Onyije et al. [24] Hu et al. [25]	Song et al. [26] Pranata et al. [20]			Song et al. [26] Pranata et al. [20] Zhao et al. [12] Cote et al. [23] Creed et al. [27]	Wang et al. [28]
Breast cancer		Wang et al. [29] Kim et al. [30] Do et al. [31]	Schmit et al. [32] Lin et al. [33]		Wang et al. [29] Gianfredi et al. [34] Zhang et al. [35] van Die et al. [36] Lamchabbek et al. [37] Lin et al. [33] Romelli et al. [38]	Shin et al. [39]
Colorectal cancer	Nordestgaard [40]	Kuo et al. [41] Wang et al. [42] Mackintosh et al. [43] Kim et al. [30] Kumar et al. [44] Oyelere et al. [45] Kunutsor et al. [46] Oyelere et al. [47] Zhang et al. [10] Romelli et al. [38]	Schmit et al. [32] Rosato et al. [48] Bradbury et al. [49] Liu et al. [50]		Wada et al. [51] Quang et al. [52] Romelli et al. [38]	Bradbury et al. [49] Nie et al. [53]
Endometrial cancer	Zhao et al. [5]	Zhao et al. [5] Nordestgaard [40] Gao et al. [54] Ye et al. [55] Crous-Bou et al. [56] Kunutsor et al. [46]	Ong et al. [57]			Zhang et al. [58]
Esophageal cancer	Masukume et al. [59] Carter et al. [60] Inoue-Choi et al. [61]	Zhao et al. [12]		Kaimila et al. [62] Eser et al. [63]	Zhao et al. [12] Qin et al. [64]	Nie et al. [53]
Gastric cancer	Martimianaki et al. [65] Liu et al. [66]	Kim et al. [30] Kim et al. [67]	Poorolajal et al. [68] Liu et al. [50] Pelucchi et al. [69]		Sasazuki et al. [70] Huang et al. [71]	Poorolajal et al. [68]
Liver cancer		Zhao et al. [5] Tanaka et al. [72] Bhurwal et al. [73] Pauwels et al. [74] Papadimitriou et al. [75] Kim et al. [30] Cai et al. [76] Chen et al. [77]			Li et al. [78] X. Zhang et al. [79] Chen et al. [77]	Tanaka et al. [72] Nie et al. [53]
Lung cancer	Zhao et al. [5] Nordestgaard [40] Seow et al. [80] Bunjaku et al. [81] Jabbari et al. [82] Kunutsor et al. [46]		Schmit et al. [32] Jin et al. [83]		Seow et al. [80] Huang et al. [84] Bunjaku et al. [81]	
Oral cancer		Xu et al. [85]			Neetha et al. [86] Kim et al. [87] Xu et al. [85]	
Ovarian cancer		Shafiei et al. [88] Huang et al. [89]			Nagle et al. [90]	Zheng et al. [91] Huang et al. [89] Gersekowski et al. [92]
Prostate cancer		Gregg et al. [93] Kunutsor et al. [46] Zhang et al. [10]	Schmit et al. [32] Sen et al. [94]		Grammatikopoulou et al. [95] Perletti et al. [96] Filippini et al. [97] Liu et al. [98]	Sen et al. [94]
Renal cancer		Rhee et al. [99]	Hashemian et al. [11] Chen et al. [100]			Hashemian et al. [11]
Skin cancer/ melanoma/non-melanoma		Oh et al. [101] Nordestgaard [40] Filippini et al. [97] Paiva et al. [102]		Ferhatosmanoglu et al. [7]	Oh et al. [101]	Filippini et al. [97]
Thyroid cancer		Shao et al. [103] Kim et al. [30]	Zamora-Ros et al. [104]	Fiore et al. [8] ***		Zamora-Ros et al. [104]
Pancreatic cancer			Liu et al. [105]			Nie et al. [53]

* Including results under specific conditions, e.g., the number of cups of coffee/tea consumed. ** Including effects on children due to maternal intake. *** Statistically insignificant increase.

**Table 2 cimb-47-00898-t002:** Regulatory effects of CGA and EGCG on miRs.

miR	Coffee/CGA Upregulation	Coffee/CGA Downregulation	Green Tea/EGCG Upregulation	Green Tea/EGCG Downregulation
miR-7-1			Wang et al. [124]	
miR-15a				Gordon et al. [125]
miR-15b			Zhang et al. [126]	
miR-16			Tsang et al. [127]	Gordon et al. [125]
miR-17		Huang et al. [128]		
miR-20a		Huang et al. [128]		Mirzaaghaei et al. [129]
miR-21		Wang et al. [130]		Siddiqui et al. [131] Fix et al. [132]
miR-21-3p	El Gizawy et al. [133]		Zhu et al. [134]	Banerjee et al. [135]
miR-22			Li et al. [136]	
miR-23b-5p			Dharshini et al. [137]	
miR-25				Gordon et al. [125] Zan et al. [138]
miR-27				Dharshini et al. [137]
miR-27a				Fix et al. [132]
miR-29			Zhu et al. [139]	
miR-29a			Zhu et al. [139]	
miR-30c	Nakayama et al. [140]			
miR-30e-3p			Wang et al. [124]	
miR-31		Zeng et al. [141] Luque-Badillo et al. [142]		
miR-33a				Baselga-Escudero et al. [143]
miR-34a		Liu et al. [144]	Toden et al. [145] Kang et al. [146] Mostafa et al. [147]	
miR-92				Dharshini et al. [137] Gordon et al. [125]
miR-92a				Mirzaaghaei et al. [129]
miR-93		Huang et al. [128]		Chakrabarti et al. [148]
miR-98-5p				Zhou et al. [149]
miR-99a			Chakrabarti et al. [148]	
miR-106b		Huang et al. [128]		Chakrabarti et al. [148]
miR-122				Baselga-Escudero et al. [143]
miR-125b			Zhu et al. [139]	Zhu et al. [139]
miR-133a		Khedr et al. [150]		Wang et al. [124]
miR-141				Gordon et al. [125]
miR-145			Toden et al. [145]	
miR-146-5p			Zhu et al. [134]	
miR-155		Zeng et al. [141] El Gizawy et al. [133]		
miR-181a			Wang et al. [124]	
miR-187-5p			Suetsugu et al. [151]	
miR-192			Zhou et al. [152]	
miR-199-5p	Wang et al. [153]			
miR-200a				Gordon et al. [125]
miR-200c			Toden et al. [145]	
miR-203			Zhu et al. [139]	Zhu et al. [139]
miR-205-3p				Li et al. [136]
miR-210			Zhu et al. [139]	
miR-215			Zhou et al. [152]	
miR-212-5p				Bhardwaj et al. [154]
miR-215			Zhou et al. [152]	
miR-218-5p			Zhu et al. [134]	Lewis et al. [155]
miR-221			Arffa et al. [156] Tsang et al. [127]	
miR-222			Wang et al. [124]	
miR-296			Lin et al. [157]	
miR-330			Siddiqui et al. [131]	
miR-483-3p				Kang et al. [158]
miR-485			Jiang et al. [159]	
miR-548m			Fix et al. [132]	
miR-720			Fix et al. [132]	
miR-1275			Shaalan et al. [160]	
miR-3176			Lee et al. [161]	Zhu et al. [134]
miR-5100			Sasaki et al. [162]	
miR-483-3p				Kang et al. [158]
let-7a			Yamada et al. [163]	
let-7b			Yamada et al. [163]	
let-7e			Yamada et al. [163]	

**Table 3 cimb-47-00898-t003:** Microarray/NGS analysis for upregulation or downregulation by EGCG/Polyphenon-60 in different cancer cells.

NGS analysis of human breast cancer MDA-MB-231 cells [135]	
Upregulated by EGCG	miR-15a-3p, miR-18a-3p, miR-30c-5p, miR-122-5p, miR-129-2-3p, miR-130a-5p, miR-138-1-3p, miR-143-3p, miR-145-5p, miR-146a-5p, miR-146a-3p, miR-150-5p, miR-155-5p, miR-184, miR-192-3p, miR-193b-5p, miR-199a-5p, miR-214-3p, miR-215-5p, miR-320a, miR-320c, miR-324-5p, miR-328-3p, miR-338-5p, miR-338-3p, miR-362-5p, miR-363-3p, miR-365b-5p, miR-365a-5p, miR-374b-5p, miR-378a-3p, miR-378c, miR-382-3p, miR-411-5p, miR-491-3p, miR-500b-3p, miR-548am-3p, miR-548ab, miR-550b-3p, miR-556-5p, miR-574-5p, miR-584-3p, miR-642a-5p, miR-664b-3p, miR-676-3p, miR-1233-3p, miR-1237-3p, miR-1249-3p, miR-1269a, miR-1272, miR-1273a, miR-1293, miR-1908-3p, miR-3074-3p, miR-3120-3p, miR-3135b, miR-3138, miR-3140-3p, miR-3145-3p, miR-3150a-5p, miR-3152-3p, miR-3155b, miR-3177-3p, miR-3184-3p, miR-3605-3p, miR-3620-5p, miR-3648, miR-3679-3p, miR-3684, miR-3909, miR-4284, miR-4436b-5p, miR-4466, miR-4485-3p, miR-4488, miR-4645-5p, miR-4661-5p, miR-4677-5p, miR-4707-5p, miR-4781-3p, miR-4791, miR-4999-5p, miR-5090, miR-5697, miR-6511a-3p, miR-6511b-5p, miR-6514-5p, miR-6515-5p, miR-6516-3p, miR-6716-3p, miR-6729-3p, miR-6739-3p, miR-6741-3p, miR-6753-5p, miR-6761-5p, miR-6769b-3p, miR-6786-3p, miR-6806-3p, miR-6811-5p, miR-6851-3p, miR-6854-3p, miR-6882-5p, miR-7111-3p, miR-7851-3p, let-7e-5p, let-7d-5p
Downregulated by EGCG	miR-17-3p, miR-19b-1-5p, miR-19b-3p, miR-21-3p, miR-26b-5p, miR-27a-3p, miR-27b-3p, miR-30c-2-3p, miR-30d-3p, miR-33a-3p, miR-33a-5p, miR-34b-5p, miR-99a-5p, miR-100-3p, miR-140-5p, miR-181b-3p, miR-190a-5p, miR-197-5p, miR-217, miR-218-1-3p, miR-296-3p, miR-301b-5p, miR-301a-3p, miR-335-5p, miR-362-3p, miR-369-3p, miR-378, miR-450a-5p, miR-489-3p, miR-508-3p, miR-516a-5p, miR-522-3p, miR-548u, miR-548ac, miR-548p, miR-551b-5p, miR-570-5p, miR-577, miR-588, miR-589-3p, miR-597-3p, miR-624-3p, miR-627-5p, miR-627-3p, miR-636, miR-653-3p, miR-708-5p, miR-762, miR-942-3p, miR-1260b, miR-1273e, miR-1273c, miR-1273h-5p, miR-1277-5p, miR-1284, miR-1538, miR-1914-3p, miR-1972, miR-1976, miR-2355-3p, miR-3064-5p, miR-3074-5p, miR-3127-5p, miR-3140-5p, miR-3149, miR-3163, miR-3190-3p, miR-3191-3p, miR-3199, miR-3529-3p, miR-3613-5p, miR-3619-3p, miR-3680-3p, miR-3690, miR-3918, miR-3944-3p, miR-4289, miR-4420, miR-4429, miR-4454, miR-4517, miR-4668-5p, miR-4684-5p, miR-4709-5p, miR-5001-3p, miR-5003-5p, miR-5006-3p, miR-5008-3p, miR-5196-3p, miR-5584-5p, miR-5584-3p, miR-5699-5p, miR-6513-5p, miR-6720-3p, miR-6726-3p, miR-6733-5p, miR-6735-5p, miR-6750-3p, miR-6783-5p, miR-6799-3p, miR-6802-3p, miR-6804-5p, miR-6814-5p, miR-6854-5p, miR-6856-3p, miR-6858-3p, miR-6871-3p, miR-6876-5p, miR-6879-3p, miR-6891-5p, miR-6895-5p, miR-7110-3p, miR-7155-5p, let-7i-3p
Microarray analysis of human nasopharyngeal carcinoma CNE2 cells [136]	
Upregulated by EGCG	miR-29b-1-5p, miR-34a, miR-210, miR-1202, miR-1207-5p, miR-1225-5p, miR-1246, miR-1915, miR-1973, miR-2861, miR-3162, miR-3196, miR-3656, miR-3665, miR-4281
Downregulated by EGCG	miR-205-3p
Next-generation sequencing NGS of human urinary transitional cell carcinoma BFTC cells [161]	
Upregulated miRNA (>2-fold change)	miR-18a-3p, miR-22-3p, miR-31-5p, miR-93-3p, miR-185-3p, miR-484, miR-642a-5p, miR-1226-3p, miR-1285-3p, miR-3139, miR-3176
Downregulated miRNA (>2-fold change)	miR-3116, miR-6724-5p
Microarray analysis of human breast cancer MCF-7 cells [132]	
Upregulated by Polyphenon-60 (>1.1-fold change)	let-7a, miR-107, miR-548m, miR-720, miR-1826, miR-1978, miR-1979
Downregulated by Polyphenon-60 (>1.1-fold change)	miR-21, miR-25, miR-26b, miR-27a, miR-27b, miR-92a, miR-125a-5p, miR-200b, miR-203, miR-342-3p, miR-454, miR-1469, miR-1977, let-7c, let-7e, let-7g

**Table 4 cimb-47-00898-t004:** miRs upregulated by EGCG or CGA and their regulatory effects on proposed molecular targets.

miR	Dose in the Culture Medium Effective on miR	Target Candidate	Effect of miR on Target * ↑, Upregulation; ↓, Downregulation
miR-16	100 μM EGCG Tsang et al. [127]	Bcl-2	↓ Yang et al. [167]
miR-22	40 μM EGCG Li et al. [136]	Wnt/β-catenin	↓ Zhang et al. [168]
miR-29	10 μg/mL EGCG Zhu et al. [139]	Wnt	↓ Tan et al. [169]
miR-34a	50 μM EGCG Chakrabarti et al. [148]	Bcl-2	↓ Yao et al. [170]
miR-99a	50 μM EGCG Chakrabarti et al. [148]	mTOR	↓ Hu et al. [171]
miR-145	100 μM EGCG Toden et al. [145]	Caspase ERK1/2 via PAK4	↑ Zhou et al. [172] ↓ Wang et al. [173]
miR-199-5a	15–120 μM CGA (not specified) Wang et al. [153]	MMP via DDR1	↓ Ravindran et al. [174]
miR-200c	100 μM EGCG Toden et al. [145]	PTEN VEGF via VEGFR2 AKT via KRAS	↑ Soubani et al. [175] ↓ Shi et al. [176] ↓ Ding et al. [177]
let-7b	10 μM EGCG Yamada et al. [163]	ERK	↓ Hameiri-Grossman et al. [178]

* Findings regarding miR targets obtained not only from experiments using CGA or EGCG, but also from those using other agents, such as drugs. AKT, AKT serine/threonine kinase 1; Bcl-2, B-cell lymphoma 2; ERK, extracellular signal-regulated kinase; KRAS, KRAS proto-oncogene; MMP, matrix metalloproteinase; mTOR, mammalian target of rapamycin; PAK4, p21-activated kinase 4; PTEN, phosphatase and tensin homologs deleted on chromosome 10; VEGF, vascular endothelial growth factor; VEGFR2, vascular endothelial growth factor receptor 2; Wnt, wingless-related integration site.

**Table 5 cimb-47-00898-t005:** miRs downregulated by EGCG, Polyphenon-60, or CGA and their regulatory effects on proposed molecular targets.

miR	Dose in the Culture Medium Effective on miR Unless Otherwise Stated	Target Candidate	Effect of miR on Target * ↑, Upregulation; ↓, Downregulation
miR-20a	25 μM CGA Huang et al. [128] 50 μg/mL EGCG Mirzaaghaei et al. [129]	p21 Huang et al. [128] PTEN Dhar et al. [179]	↓ Huang et al. [128] ↑ Dhar et al. [179]
miR-21	5 mg/kg CGA in mice Wang et al. [130] 10 μg/mL Polyphenon-60 Fix et al. [132]	PTEN CUR: Zhang et al. [180] p21 Zaman et al. [181] Bcl-2 Liu et al. [182] NF-κB Li et al. [183]	↓ Zhang et al. [180] ↓ Zaman et al. [181] ↑ Liu et al. [182] ↑ Li et al. [183]
miR-25	1 μM EGCG Gordon et al. [125] 10 μg/mL Polyphenon-60 Fix et al. [132]	p53 Gordon et al. [125] ERK1/2 via KLF4 Ding et al. [177]	↓ Gordon et al. [125] ↑ Ding et al. [177]
miR-27a	10 μg/mL Polyphenon-60 Fix et al. [132]	NFκB Shi et al. [184] EGFR, Bcl-2, NF-κB Gandhy et al. [185]	↑ Shi et al. [184] ↑ Gandhy et al. [185]
miR-92	1 μM EGCG Gordon et al. [125]	p53 Gordon et al. [125]	↓ Gordon et al. [125]
miR-93	50 μM CGA Huang et al. [128] 50 μM EGCG Chakrabarti et al. [148]	Caspases Chakrabarti et al. [186] p21 Huang et al. [128]	↓ Chakrabarti et al. [186] ↓ Huang et al. [128]
miR-106b	25 μM CGA Huang et al. [128] 50 μM EGCG Chakrabarti et al. [148]	p21 Huang et al. [128] PTEN Dhar et al. [187]	↓ Huang et al. [128] ↓ Dhar et al. [187]
miR-125b	40 μM EGCG Zhu et al. [139]	NF-κB [188]	↑ Song et al. [188]
miR-133a	Green coffee extract containing CGA equivalent to 400 mg in RCT Khedr et al. [150] 50 mg/kg EGCG in rats Zhou et al. [189]	NF-κB [190]	↑ Wang et al. [190]
miR-141	1 μM EGCG [125]	p53 Gordon et al. [125]	↓ Gordon et al. [125]
miR-155	31.25 μM CGA [125]	NF-κB CGA: Zeng et al. [141] PTEN de la Parra et al. [191]	↑ Zeng et al. [141] ↓ de la Parra et al. [191]
miR-221 Allegri et al. [77]	50 μM EGCG [192]	PTEN Sarkar et al. [193] MMP2 Zhang et al. [194]	↓ Sarkar et al. [193] ↑ Zhang et al. [194]
483-3p Kang et al.	30 μM EGCG [158]	VEGF via OGT Kim et al. [195]	↑ Kim et al. [195]

* Findings regarding miR targets obtained not only from experiments using CGA, EGCG, or Polyphenon-60, but also from those using other agents, such as drugs. Bcl-2, B-cell lymphoma 2; EGFR, epidermal growth factor receptor; ERK, extracellular signal-regulated kinase; MMP, matrix metalloproteinase; OGT, O-GlcNAc transferase; PTEN, phosphatase and tensin homologs deleted on chromosome 10; VEGF, vascular endothelial growth factor.

## Data Availability

No new data were created or analyzed in this study. Data sharing is not applicable to this article.

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
