# Peer review of "Regulatory Effects of Coffee/Chlorogenic Acid and Tea/Epigallocatechin-3-O-Gallate on microRNA in Association with Their Anticancer Activity"

_cimb, 2025, doi:10.3390/cimb47110898_

Round 1
Reviewer 1 Report
Comments and Suggestions for Authors
This review covers human studies on coffee(CGA) and tea (EGCG), showing how these compounds may affect cancer-related microRNAs through ROS pathways. It's well organized but needs clearer explanations of dose relevance, and a distinction between direct and indirect evidence, and also a more balanced view on debated topics like dietry miRNAs, and a note on safety at high doses.
Major points-
- Many in vitro studies cited in section 3 and 5 use supra-physiological concentrations of EGCG and CGA to demonstrate effects on DNA methylation, histone acetylation, miRNAs, and ROS pathways. however, in humans, plasma levels after normal tea or coffee consumption are much lower (EGCG : Lee et al., 2002, Cancer Epidemiol Biomarkers Prev,; CGA: Stalmach et al., 2014, Mol Nutr Food Res; Del Rio et al., 2010, Mol Nutr Food Res).Therefore the mechanistic effects observed at high concentraions should be considered hypothesis-generating rather than directly physiologically relevant. To strengthen the review, the authors could add a brief paragraph clarifying this or include a small table comparing experimental versus achievable human plasma levels.
- To improve clarity, please indicate which findings are direct, indirect or predicted. Table 4 could include a column marking evidence type for each miRNA target.
- Section 2.2 & conclusion: A short note should have been added to highlight that habitual tea consumption is generally safe but high- dose EGCG supplements (≥800 mg/day) may cause liver toxicity (Hu et al., 2018).
Minor points-
- Rephrase statement such as "CGA and EGCG exert anticancer effects" to more cautious wording like "may contribute to anticancer pathways.
- Suggested rewording for section 7 conclusion: EGCG and CGA may modulate anticancer pathways via ROS signaling and miRNA regulation. However, most evidence remains in vitro or indirect: further human studies are needed before firm conclusions can be drawn.
The review is valuable and nearly ready for publication. Minor clarifications reagarding dose relevance, evidence, validation, and a balanced discussion of dietry miRNAs and safety will further improve usefullness.
Author Response
Responses to Reviewers’ comments
Reviewer 1
Comments and Suggestions for Authors
This review covers human studies on coffee (CGA) and tea (EGCG), showing how these compounds may affect cancer-related microRNAs through ROS pathways. It's well organized but needs clearer explanations of dose relevance, and a distinction between direct and indirect evidence, and also a more balanced view on debated topics like dietary miRNAs, and a note on safety at high doses.
Major points-
- Many in vitro studies cited in section 3 and 5 use supra-physiological concentrations of EGCG and CGA to demonstrate effects on DNA methylation, histone acetylation, miRNAs, and ROS pathways. however, in humans, plasma levels after normal tea or coffee consumption are much lower (EGCG : Lee et al., 2002, Cancer Epidemiol Biomarkers Prev,; CGA: Stalmach et al., 2014, Mol Nutr Food Res; Del Rio et al., 2010, Mol Nutr Food Res). Therefore, the mechanistic effects observed at high concentrations should be considered hypothesis-generating rather than directly physiologically relevant. To strengthen the review, the authors could add a brief paragraph clarifying this or include a small table comparing experimental versus achievable human plasma levels. To improve clarity, please indicate which findings are direct, indirect or predicted. Table 4 could include a column marking evidence type for each miRNA target.
Author response:
Thank you for these constructive comments. We agree that several studies may not have physiological relevance. However, it appears difficult to categorically determine whether the obtained results hold direct, indirect, or hypothetical physiological significance. Therefore, in Table 4, we added data on the concentration used in the individual experiment to give information on this issue. For clarity of presentation, original Table 4 has been divided into Table 4 and Table 5.
- Section 2.2 & conclusion: A short note should have been added to highlight that habitual tea consumption is generally safe but high- dose EGCG supplements (≥ 800 mg/day) may cause liver toxicity (Hu et al., 2018).
Author response:
Thank you for this comment. A description “habitual tea consumption is generally safe (e.g. 704 mg/day in beverage) but higher dose EGCG (≥800 mg/day) may cause liver toxicity [PMID: 29580974].” was added in Section 2.2.
Minor points-
- Rephrase statement such as "CGA and EGCG exert anticancer effects" to more cautious wording like "may contribute to anticancer pathways.
Author response:
Thanks to this comment, “exert anticancer effects” in the legend for Figure 2 has been changed to “may contribute to anticancer pathways”.
- Suggested rewording for section 7 conclusion: EGCG and CGA may modulate anticancer pathways via ROS signaling and miRNA regulation. However, most evidence remains in vitro or indirect: further human studies are needed before firm conclusions can be drawn. The review is valuable and nearly ready for publication. Minor clarifications regarding dose relevance, evidence, validation, and a balanced discussion of dietary miRNAs and safety will further improve usefulness.
Author response:
In the section of Conclusion, a sentence has been added: “In addition, most evidence described in this review remains in vitro or indirect, and future human studies would be needed before firm conclusions can be drawn.”
Reviewer 2 Report
Comments and Suggestions for Authors
The manuscript entitled “Regulatory effects of coffee/chlorogenic acid and tea/epigallocatechin-3-O-gallate on microRNA in association with their anticancer activity” by Mamoru Isemura et al. provides an overview of the link between dietary polyphenols (CGA and EGCG) and cancer prevention via epigenetic modification through microRNAs. While the topic is relevant, the manuscript in its current form requires major revisions before it can be considered for publication.
However, I do have some major concerns about this work:
1. To what extent is it appropriate to discuss the biological activity of such heterogenic beverages as coffee and tea using CGA and EGCG as the only acting compounds? Both beverages contain a wide variety of active compounds, which can exert effects individually or in combination. For instance, coffee consists of anticancer biologically active molecules like caffeine, diterpenes (cafestol, kahweol), melanoidins, trigonelline, etc. The only way to prove the main point of the article—the specific biological activity of coffee/green tea based on the CGA/EGCG—is to extract them from the crude coffee and then show that without them, the raw material does not have such activity.
2. Is it correct to discuss the influence of CGA on the population health without discussing the extraction procedure? The extraction process may be more significant than the mere appearance or concentration of active molecules in coffee beans. The roasting procedure can also lead to the formation of biologically active compounds through thermal and chemical reactions that may possess pro- or anticancer activity.
3. Moreover, one of coffee's components is acrylamide, which is classified as a probable human carcinogen by the International Agency for Research on Cancer. Some research suggests that the antioxidant compounds in coffee may counteract the potential harmful effects of acrylamide, but do they? Is it possible to assess the extent of the benefits? This should be discussed and clarified.
4. At the same time, Section 2.1 lists numerous confounding factors (temperature, alcohol, genetics, etc.), which inherently reduce the certainty of associating coffee/tea consumption directly with CGA/EGCG effects.
5. A major uncertainty is highlighted in Section 3 (Lines 177-187): Table 3 is dominated by microarray/NGS data, much of which remains unconfirmed by qRT-PCR. NGS platforms often generate high numbers of findings that may not be biologically robust or reproducible (false positives). The authors temper this by noting miR-34a and let-7a confirmation, but the bulk of the 150+ miRs listed in Table 3 are based on less validated data.
6. Table 1 shows that for the same cancer type (e.g., bladder cancer), one citation shows up-regulation of risk by CGA, while another shows down-regulation. The review acknowledges this inconsistency but does not answer the question “why?”
7. Coffee consumption generally shows no link to increased or decreased cancer risk. Please utilize global statistics to create a correlation plot between the incidence of cancer cases and coffee consumption rates.
8. Lack of molecular mechanisms of CGA and EGCG downregulate miRs.
9. It needs to be discussed: the dose-activity dependency of CGA and EGCG from beverages.
Author Response
Responses to Reviewers’ comments
Reviewer 2
Comments and Suggestions for Authors
The manuscript entitled “Regulatory effects of coffee/chlorogenic acid and tea/epigallocatechin-3-O-gallate on microRNA in association with their anticancer activity” by Mamoru Isemura et al. provides an overview of the link between dietary polyphenols (CGA and EGCG) and cancer prevention via epigenetic modification through microRNAs. While the topic is relevant, the manuscript in its current form requires major revisions before it can be considered for publication.
However, I do have some major concerns about this work:
- To what extent is it appropriate to discuss the biological activity of such heterogenic beverages as coffee and tea using CGA and EGCG as the only acting compounds? Both beverages contain a wide variety of active compounds, which can exert effects individually or in combination. For instance, coffee consists of anticancer biologically active molecules like caffeine, diterpenes (cafestol, kahweol), melanoidins, trigonelline, etc. The only way to prove the main point of the article—the specific biological activity of coffee/green tea based on the CGA/EGCG—is to extract them from the crude coffee and then show that without them, the raw material does not have such activity.
Author response:
Thank you for your precious comments. As pointed out, it appears difficult to conclude what components are responsible for the anticancer effects of coffee and tea, which contain a wide variety of active compounds. One method for identifying contributing components is indeed to use extracts that do not contain candidate compounds. Another approach would be to use 100 % pure preparations of CGA or EGCG. We believe that the in vitro results described in this review can be attributed to CGA or EGCG, as their purity was sufficiently high, although not absolutely 100%.
- Is it correct to discuss the influence of CGA on the population health without discussing the extraction procedure? The extraction process may be more significant than the mere appearance or concentration of active molecules in coffee beans. The roasting procedure can also lead to the formation of biologically active compounds through thermal and chemical reactions that may possess pro- or anticancer activity.
Author response:
Thank you for your comment. We agree with you that many factors influence the results of an epidemiological study, including extraction methods, roasting procedures, coffee species, and serving temperature. However, it appears very difficult to discuss the reported study results by incorporating such complex confounding factors.
- Moreover, one of coffee's components is acrylamide, which is classified as a probable human carcinogen by the International Agency for Research on Cancer. Some research suggests that the antioxidant compounds in coffee may counteract the potential harmful effects of acrylamide, but do they? Is it possible to assess the extent of the benefits? This should be discussed and clarified.
- At the same time, Section 2.1 lists numerous confounding factors (temperature, alcohol, genetics, etc.), which inherently reduce the certainty of associating coffee/tea consumption directly with CGA/EGCG effects.
Author response:
Thank you for your comprehensive comment. It is possible that acrylamide in coffee may compensate for the CGA’s benefits. Regarding this issue, in a recent review article, Burhan BaÅŸaran et al. [PMID: 36673439] have discussed based on positive, negative, and no relationships between dietary acrylamide and cancer risk. These authors have concluded that examining the relationship should be planned to include more people and foods in order to obtain more reliable results, suggesting the need to correct dietary acrylamide contents as a confounding factor. Therefore, we added acrylamide as one of the significant confounding factors.
- A major uncertainty is highlighted in Section 3 (Lines 177-187): Table 3 is dominated by microarray/NGS data, much of which remains unconfirmed by qRT-PCR. NGS platforms often generate high numbers of findings that may not be biologically robust or reproducible (false positives). The authors temper this by noting miR-34a and let-7a confirmation, but the bulk of the 150+ miRs listed in Table 3 are based on less validated data.
Author response:
Thank you for your comprehensive comment. According to this comment, we added a description: “Thus, NGS platforms provide a large number of findings, but these may not be biologically robust and may lack reproducibility. Therefore, validation of these findings would be necessary.”
- Table 1 shows that for the same cancer type (e.g., bladder cancer), one citation shows up-regulation of risk by CGA, while another shows down-regulation. The review acknowledges this inconsistency but does not answer the question “why?”
Author response:
Thank you for this comment. It is difficult to answer the question “Why”. The present authors can only point out that the possible differences may be due to various confounding factors as listed.
- Coffee consumption generally shows no link to increased or decreased cancer risk. Please utilize global statistics to create a correlation plot between the incidence of cancer cases and coffee consumption rates.
Author response:
We appreciate the reviewer’s valuable suggestion. Following this comment, we have created a correlation plot (Figure 2) using country-level coffee consumption data from the FAO/European Coffee Report 2023–2024 (https://www.ecf-coffee.org/european-coffee-report-2023-2024/) and age-standardized cancer incidence data (excluding non-melanoma skin cancer) from WHO/IARC GLOBOCAN 2022 (https://www.wcrf.org/preventing-cancer/cancer-statistics/global-cancer-data-by-country/#global-cancer-incidence-both-sexes). The analysis revealed a weak positive association between per-capita coffee consumption and total cancer incidence, indicating that coffee intake alone does not explain global variation in cancer occurrence. This ecological result is subject to multiple confounders such as smoking, alcohol, diet, and healthcare accessibility. A meta-analysis of 59 cohort studies reported that higher coffee intake was associated with a reduced overall cancer risk (RR = 0.87, 95% CI: 0.82–0.92), and particularly lower risks of liver, endometrial, and prostate cancers, whereas an increased risk was observed for lung cancer (RR = 2.18, 95% CI: 1.26–3.78) [PMID: 21406107]. A more recent dose–response meta-analysis reported a significant positive association between coffee consumption and lung cancer risk (RR = 1.28, 95% CI: 1.12–1.47) [PMID: 38951141]. An umbrella review of meta-analyses [PMID: 38951141] concluded that coffee consumption shows highly suggestive inverse associations with liver and endometrial cancer risks, indicating potential protective effects rather than overall carcinogenicity. These findings together indicate that there is no consistent global relationship between coffee consumption and total cancer incidence, although effects differ depending on cancer type.
According to the comment, a revised version provides Figure 2 to highlight that coffee consumption may have beneficial effects on certain types of cancer, and also includes citations of studies to show beneficial effects on certain types of cancer. A brief summary sentence has also been added in the Conclusion: Although a global correlation plot shows a trend of increased risk of total cancer cases with coffee consumption, several epidemiological studies have suggested their inverse associations in some specific cancer types.
- Lack of molecular mechanisms of CGA and EGCG downregulate miRs.
Author response:
Thank you for this comprehensive comment. Molecular mechanisms of CGA and EGCG for up/down-regulation of miRs are largely unknown, although one of them is likely to be DNA modulation by affecting DNMT activity and histone acetylase, as discussed in the Text. A variety of factors are involved in the biogeneration of miRs, but research to give information on how EGCG or CGA can regulate miRs is largely unavailable. One possibility is the EGCG’s ability to bind DNA and RNA [PMID: 16641087], which would be able to regulate the generation of miRs. This aspect is described in a revised version: These are the examples to explain how CGA and EGCG can modulate miR expression. While EGCG’s ability to bind DNA and RNA [PMID: 16641087] represents another possible mechanism, further studies are required to elucidate how CGA and/or EGCG can regulate miR generation.
- It needs to be discussed: the dose-activity dependency of CGA and EGCG from beverages.
Author response:
Thank you for this comment. Most of the in vitro data cited are those based on the findings from the dose-dependent activity of CGA and EGCG. For example, CGA’s down-regulation of miR-20a was examined at 25 and 50 μM [PMID: 31660066], and expression of miR-483-3p was examined using EGCG at 10, 20, 30, 40, 50, and 60 μM [PMID: 33900350].
Round 2
Reviewer 2 Report
Comments and Suggestions for Authors
The manuscript entitled “Regulatory effects of coffee/chlorogenic acid and tea/epigallocatechin-3-O-gallate on microRNA in association with their anticancer activity” by Mamoru Isemura et al. provides an overview of the link between dietary polyphenols (CGA and EGCG) and cancer prevention via epigenetic modification through microRNAs. While the topic is relevant, the manuscript in its current form requires major revisions before it can be considered for publication.
1. Authors’ statement: “We believe that the in vitro results described in this review can be attributed to CGA or EGCG, as their purity was sufficiently high, although not absolutely 100%.” does not answer my concern. The only way is to discuss this limitation in the Discussion section.
2. Same thing for: “Thank you for your comment. We agree with you that many factors influence the results of an epidemiological study, including extraction methods, roasting procedures, coffee species, and serving temperature. However, it appears very difficult to discuss the reported study results by incorporating such complex confounding factors.” Add it as a limitation to the Discussion section. It is always beneficial to understand study limitations.
6. If there is no certain answer to the question 'why', then you can discuss and propose your hypothesis. For instance, CGA shows up- or downregulation depending on other demographic parameters, which have not been taken into account in these studies. Or perhaps different concentrations of active components in different studies lead to up- or down-regulation?
8. It would be beneficial to add hypothetical molecular mechanisms into the text.
9. Please add this information to the text. Now there is a problem to understand – where and what concentration of CGA and EGCG has been used?
Author Response
Comments and Suggestions for Authors
The manuscript entitled “Regulatory effects of coffee/chlorogenic acid and tea/epigallocatechin-3-O-gallate on microRNA in association with their anticancer activity” by Mamoru Isemura et al. provides an overview of the link between dietary polyphenols (CGA and EGCG) and cancer prevention via epigenetic modification through microRNAs. While the topic is relevant, the manuscript in its current form requires major revisions before it can be considered for publication.
We appreciate the Round 2 comments provided by Reviewer 2. After our previous revision, we re‑examined the manuscript and have revised and supplemented it to address the concerns raised. Below we outline our responses to each point. We wish to emphasize that the corresponding changes have already been incorporated into the previous revised manuscript.
- Authors’ statement: “We believe that the in vitro results described in this review can be attributed to CGA or EGCG, as their purity was sufficiently high, although not absolutely 100%.” does not answer my concern. The only way is to discuss this limitation in the Discussion section.
As you rightly point out, coffee and green tea contain many constituents besides CGA and EGCG, including caffeine, diterpenes, melanoidins, and acrylamide. We have clarified in the Conclusion that there are limits to attributing the activity of extracts or whole beverages to a single polyphenol (L330-335). When summarizing in vitro evidence in this review we now explicitly note that the cited data were obtained using high‑purity CGA or EGCG preparations and state that future work should compare these preparations with extracts lacking the compounds or use 100 % purified compounds. In this way, we acknowledge the current limitations and highlight the need for further verification.
- Same thing for: “Thank you for your comment. We agree with you that many factors influence the results of an epidemiological study, including extraction methods, roasting procedures, coffee species, and serving temperature. However, it appears very difficult to discuss the reported study results by incorporating such complex confounding factors.” Add it as a limitation to the Discussion section. It is always beneficial to understand study limitations.
The revised manuscript now lists potential confounders in epidemiological studies, including extraction method, roasting condition, plant variety, acrylamide content, serving temperature, alcohol and smoking history, genetic diversity, dietary habits, and gut microbiota, and explains in Section 2 (L121) that these factors may contribute to the variability in observed results. We have added to the Conclusion that few studies control for these variables, so results should be interpreted with caution, and future work requires detailed data collection and analysis. We believe this revision makes the limitations of current evidence clearer to readers.
- If there is no certain answer to the question 'why', then you can discuss and propose your hypothesis. For instance, CGA shows up- or downregulation depending on other demographic parameters, which have not been taken into account in these studies. Or perhaps different concentrations of active components in different studies lead to up- or down-regulation?
We have expanded the discussion (L335-340) of why opposite results have been reported for the same cancer type. We note that demographic factors (age, sex, smoking status, and genetic background) differ among study populations and that differences in the concentration of active constituents in the beverages may also influence outcomes. We also point out that intake amount, serving temperature, and concomitant consumption of alcohol or other foods may act as confounders. The Discussion now proposes that these factors might underlie the inconsistent findings and recommends that future studies examine them in detail.
- It would be beneficial to add hypothetical molecular mechanisms into the text.
The revised manuscript now describes several possible mechanisms by which CGA or EGCG could modulate miRNA expression. These include (i) changes in DNA methylation patterns through inhibition of DNA methyltransferase (DNMT) activity (L172~); (ii) increased histone acetylation via inhibition of histone deacetylases (L179~); and (iii) EGCG’s ability to bind directly to DNA or RNA, which could affect miRNA biogenesis (L184~). These are hypothetical mechanisms based on current reports, and we emphasize that further research is needed to test them (L186-189).
- Please add this information to the text. Now there is a problem to understand – where and what concentration of CGA and EGCG has been used?
In response, we now explicitly state the concentrations used in the cited studies throughout the manuscript. For example, we note that suppression of miR‑20a by CGA was observed at 25 and 50 µM and that down‑regulation of miR‑483‑3p by EGCG was reported across a range of 10–60 µM. Such concentration information, together with citations, is now provided in the text (L168-171) so that readers can track experimental conditions.
We carefully reviewed our manuscript in light of the Round 2 comments and added explanations and discussion where information was lacking. In particular, we have clarified the limitations of attributing effects to single compounds, considered confounding factors, proposed hypothetical mechanisms, and provided concentration details. We believe these revisions make the current evidence and its limitations clearer and help to guide future research.
Round 3
Reviewer 2 Report
Comments and Suggestions for Authors
The authors took into account all the comments